# AI-Assisted identification of sex-specific patterns in diabetic retinopathy using retinal fundus images

Parsa Delavari [1,2], Gulcenur Ozturan[1], Eduardo V. Navajas[1], Ozgur Yilmaz[3], Ipek Oruc [1,2]*

1 Ophthalmology and Visual Sciences, UBC, Vancouver, British Columbia, Canada, 2 Neuroscience, UBC, Vancouver, British Columbia, Canada, 3 Mathematics, UBC, Vancouver, British Columbia, Canada

* ipor@mail.ubc.ca

**Data availability statement:** Data cannot be shared publicly because data are owned by a third party and we do not have permission to share the data. The dataset used in this study

## Abstract

Diabetic retinopathy (DR) is a microvascular complication of diabetes that can lead to blindness if left untreated. Regular monitoring is crucial for detecting early signs of referable DR, and the progression to moderate to severe non-proliferative DR, proliferative DR (PDR), and macular edema (ME), the most common cause of vision loss in DR. Currently, aside from considerations during pregnancy, sex is not factored into DR diagnosis, management or treatment. Here we examine whether DR manifests differently in male and female patients, using a dataset of retinal images and leveraging convolutional neural networks (CNN) integrated with explainable artificial intelligence (AI) techniques. To minimize confounding variables, we curated 2,967 fundus images from a larger dataset of DR patients acquired from EyePACS, matching male and female groups for age, ethnicity, severity of DR, and hemoglobin A1C levels. Next, we fine-tuned two pre-trained VGG16 models—one trained on the ImageNet dataset and another on a sex classification task using healthy fundus images—achieving AUC scores of 0.72 and 0.75, respectively, both significantly above chance level. To uncover how these models distinguish between male and female retinas, we used the Guided Grad-CAM technique to generate saliency maps, highlighting critical retinal regions for correct classification. Saliency maps showed CNNs focused on different retinal regions by sex: the macula in females, and the optic disc and peripheral vasculature along the arcades in males. This pattern differed noticeably from the saliency maps generated by CNNs trained on healthy eyes. These findings raise the hypothesis that DR may manifest differently by sex, with women potentially at higher risk for developing ME, as opposed to men who may be at greater risk for PDR.

## Introduction

Men and women are affected differently across a range of medical conditions from cardiovascular to mental health disorders [1–5]. Although clinical manifestations and presentation of many diseases vary between men and women [6–8] key differences remain under-recognized

was obtained from EyePACS, LLC under a standard licensing agreement. We confirm that we did not receive any special access privileges that other researchers would not have. Researchers can request a license to access the data by contacting EyePACS at contact@eyepacs.org, or at 1-800-228-6144, or visiting their website https://www.eyepacs.com. More information on licensing and access conditions can be obtained by contacting EyePACS. The code utilized to generate the results presented in this manuscript is available in the following GitHub repository: https://github.com/Fundus-AI/DR_sex_difference.

**Funding:** (OY) NSERC Discovery Grant (22R82411) (OY) Pacific Institute for the Mathematical Sciences (PIMS) CRG 33 (IO) NSERC Discovery Grant (RGPIN-2019-05554) (IO) NSERC Accelerator Supplement (RGPAS-2019-00026) (IO & OY) UBC DSI Grant (no number) (IO) UBC Faculty of Science STAIR grant (IO & OY) UBC DMCBH Kickstart grant (IO & OY) UBC Health VPR HiFi grant. The sponsors or funders did not play any role in the study design, data collection and analysis, decision to publish, or preparation of the manuscript.

**Competing interests:** The authors have declared that no competing interests exist.

due to a paucity of research that systematically takes sex and gender into consideration [9,10]. Biases in diagnosis and treatment contribute to poorer outcomes across all sexes and genders, including men, women, and gender-diverse individuals. For example, osteoporosis is often underdiagnosed and undertreated in men, significantly impacting their morbidity and quality of life [3]. Similarly, while diabetes is more prevalent among men [11], women are diagnosed at an older age, greater disease severity, and higher body fat mass than men [12–15]. In addition, growing evidence suggests that diabetic complications manifest differently in women and men [16,17]. For instance, some studies indicate that women and girls may be at greater risk of developing diabetic kidney disease compared to age-matched men and boys [18,19]. Computational modeling predictions further support sex-specific differences in diabetes-induced changes in kidney function, highlighting distinct physiological responses in males and females [20]. Elucidating specific fine-grained diagnostic markers for men and women would enhance diagnostic precision and enable earlier detection of disease across diverse populations.

Diabetic retinopathy (DR), a microvascular complication of diabetes and a leading cause of blindness, affected an estimated 103 million people worldwide in 2020, with projections rising to 160 million by 2045 [21]. Delays to the diagnosis of DR poses significant risks, including progression to vision-threatening stages and systemic complications. Identifying novel sex-specific markers would likely enhance the diagnostic accuracy of DR, enabling earlier and more precise detection of the disease. There is some evidence in the literature that sex and gender play a role in how this condition develops, manifests and progresses [22–26]. However, aside from considerations during pregnancy, sex is not factored into the diagnosis, management or treatment of DR. This lack of sex- and gender-specific medical guidelines for DR highlights the need for more research into the underlying mechanisms of these differences to inform more precise diagnostic and treatment strategies. Here, we investigate sex-specific manifestations of DR using artificial intelligence to analyze retinal fundus images.

To achieve this, we utilize a comprehensive dataset of retinal images to compare male and female patients diagnosed with DR. We employ a recently developed methodological pipeline that integrates convolutional neural networks (CNNs) with explainable artificial intelligence (AI) techniques to analyze retinal fundus images [27]. AI applications have become increasingly popular in medical image analysis, particularly in ophthalmology, where machine learning models have been successfully used to detect various eye diseases, including DR, with high sensitivity and precision [28–34]. However, the broader adoption of deep learning in medicine is challenged by the "black box" nature of these models. While CNNs are powerful, they often operate in a way that is not transparent, making it difficult to understand how they arrive at their decisions. This lack of clarity poses a significant barrier to integrating AI into clinical settings, where decision-making transparency is critical. Explainable AI methods, such as post-hoc interpretability algorithms, offer a possible solution by providing visualizations that highlight important areas of medical images, like the heat maps generated by the Grad-CAM algorithm [35–37].

The rationale of the present work is as follows: Previous studies have shown that CNNs can successfully classify fundus images based on sex [38–40] and have identified several retinal markers that differ between healthy male and female eyes [27]. Here, we investigate whether a CNN model can be trained to identify sex in retinal fundus images of DR patients. If sex differences in the manifestation of DR exist, the model would likely extract and utilize these differences, in addition to any known markers of sex. Thus, we proceed by first training a CNN model to classify the sex of patients based on fundus images from those with DR, and then applying the Guided Grad-CAM technique to explore sex differences in these retinas, aiming to identify distinct manifestations of DR in males and females.

## Materials and methods

### Fundus dataset

We used a private EyePACS dataset [41] (access date: February 18, 2022), which contains approximately 90,000 fundus images, of which 9,944 were labeled with a DR diagnosis. After matching male and female images for age, ethnicity, severity of DR, and hemoglobin A1c (HbA1c) level, and including only images of good and excellent quality, we obtained a subset of 2,967 DR-labeled images, of which 1,491 were from female patients and 1,476 were from male patients. The DR levels include mild nonproliferative, moderate nonproliferative, severe nonproliferative, and proliferative DR. This subset was partitioned into training, validation, and testing sets containing 2,071, 448, and 448 images, respectively. The composition of this dataset in terms of age, sex, ethnicity, severity of DR, and HbA1c level is shown in Table 1. The dataset composition, presented separately for the training, validation, and test sets, is provided in S1–S3 Tables. Matching of male and female groups for age, HbA1c, ethnicity, and severity of DR was done to prevent the model from relying on these potentially confounding variables when classifying a patient's sex. After matching, the effect size of differences in age and HbA1c between males and females was 0.16 and 0.01, respectively, while the total variation distance in ethnicity and DR severity between the two groups was 0.09 and 0.08, respectively. Each fundus image was cropped into a square format by detecting the circular contour of the fundus and centring it within a square with equal height and width. The dataset did not contain any identifiable information.

### CNN architecture

Following our previous work [27], we used the VGG16 architecture [42] and fine-tuned two pre-trained models: (1) one pre-trained on the ImageNet dataset [43], and (2) another pre-trained on a sex classification task using fundus images from healthy individuals (no DR). The original VGG16 model, designed for 1000 output classes for the ImageNet

**Table 1. Composition of the CNN Development set. The CNN Development Set were balanced with respect to the size of the female vs. male sets, which were matched for age, ethnicity, severity of DR, HbA1c level, and years with diabetes. NPDR: Non-proliferative DR.**

| CNN Development Set | | Female | Male | Total |
|---|---|---|---|---|
| N | | 1491 | 1476 | 2967 |
| Age (M ± SD) | | 52.35 ± 11.06 | 50.68 ± 10.22 | 51.352 ± 10.69 |
| Ethnicity (N) | | | | |
| | Latin American | 1118 | 1011 | 2129 |
| | Caucasian | 136 | 240 | 376 |
| | Multi-racial | 79 | 58 | 137 |
| | Asian | 52 | 53 | 105 |
| | African Descent | 46 | 60 | 106 |
| | Other | 34 | 26 | 60 |
| | Native American | 14 | 16 | 30 |
| | Indian Subcontinent Origin | 7 | 10 | 17 |
| | Unknown | 5 | 2 | 7 |
| Severity of DR (N) | | | | |
| | Mild NPDR | 679 | 568 | 1247 |
| | Moderate NPDR | 696 | 794 | 1490 |
| | Severe NPDR | 57 | 73 | 130 |
| | Proliferative NPDR | 59 | 41 | 100 |
| HbA1c (M ± SD) | | 8.95 ± 3.06 | 8.92 ± 3.92 | 8.94 ± 3.52 |

classification contest, was modified for our binary sex classification task by replacing the final fully connected (FC) layer with a randomly initialized FC layer. This new layer had 4096 input features (corresponding to the VGG model's output features) and two output classes (male and female).

## Training procedure

Our training approach combined transfer learning with fine-tuning. For the first two epochs, we kept the network's weights frozen, allowing only the newly added classifier layer to learn the task. This practice prevents the initial random weights of the FC layer from steering the network parameters in undesired directions. After these initial two epochs, once the classifier was partially trained, we unfroze the network's weights and allowed them to be updated over the next 100 epochs. Hyperparameters were optimized by testing various combinations and selecting those with the best validation performance. A summary of the hyperparameters used for model architecture, training and evaluation is provided in Table 2.

## Data augmentation and transforms

During CNN training, we implemented data augmentation techniques to prevent the model from merely memorizing image-label pairs. We first applied random rotations to the images, ranging uniformly from -10 to +10 degrees. We also introduced an innovative approach specifically tailored to fundus image datasets. Since the left and right retinas are mirror images with approximate symmetry along the vertical axis, they introduce significant image-level variation (left vs. right). However, this source of variation is not relevant to the sex classification task. To address this, we horizontally flipped all right-eye images so that they resemble left-eye fundus images. This horizontal flipping transformation reduces task-irrelevant image variance in the dataset, improving model performance in sex classification [27]. The rationale for this improvement is that horizontal flipping ensures the model encounters the same

**Table 2. Summary of the hyperparameters used for training and evaluating the model.**

| Training Hyperparameters | |
|---|---|
| **Optimizer** | |
| **method** | Adam |
| **batch size** | 128 |
| **number of epochs** | 102 |
| **initial learning rate** | 0.0003 |
| **learning rate annealing** | 0.5 every 20 epochs |
| **Criterion** | |
| **loss function** | binary cross-entropy |
| **class weights (Female, Male)** | (0.463, 0.537) |
| **Network** | |
| **architecture** | VGG16 |
| **input image resolution** | 224×224 |
| **number of features (hidden layer)** | 4096 |
| **number of output classes** | 2 |
| **Training Transforms** | |
| **random rotation** | $\theta \sim Uniform(-10°, +10°)$ |
| **resize** | 224×224 |
| **Validation Transforms** | |
| **resize** | 224× 224 |

anatomical retinal features in consistent locations (e.g., optic disc on the left, and fovea on the right), facilitating more efficient feature learning.

## Model evaluation

We used the validation set to assess the model's performance and tune the hyperparameters based on the AUC score. In addition to AUC, we monitored accuracy and binary cross-entropy (BCE) loss on both training and validation sets throughout the epochs to track training progress. To determine the best-performing model, we selected the epoch with the highest validation AUC score and saved the model's weights at this point as "the best model's weights." Subsequently, we reloaded this best model to evaluate and report its performance on an unseen test set.

## Generation of saliency maps (Grad-CAM)

To generate saliency maps, we used the Grad-CAM (Gradient-weighted Class Activation Mapping) technique, introduced by Selvaraju et al. [37]. The input images were fed into the trained CNN for a forward pass, during which we saved the predicted labels. The model's output was one-hot encoded, designating the predicted class as one and the other class as zero. The Grad-CAM saliency map was generated by backpropagating the gradient of the predicted label to the last convolutional layer. Simultaneously, we computed the Guided Back-propagation map through a deconvolutional network designed as the inverse of the trained model (see [36] for details). Finally, the outputs of these two methods were combined through pixel-wise multiplication to generate the Guided Grad-CAM saliency maps. In the saliency maps, each colour channel reflects the attention given to its corresponding colour (middle row in Figs 1 and 2). To provide a clearer view of the overall attention distribution across the image, we also converted the color saliency maps into amplitude-only versions by aggregating the three colour channels. These amplitude maps are colour-coded and displayed in the right column of Figs 1 and 2.

## Results

### Sex classification

The models were tested on the unseen test set, and the performance results are reported in Table 3. To determine the significance of the results and calculate p-values, we compared performance metrics achieved by each model to chance-level performance using a non-parametric t-test with bootstrapping. The ImageNet pre-trained model achieved an average test AUC of 0.72 and test accuracy of 0.66, both significantly above the chance level (*p*-value <0.001). The model pretrained on sex classification showed improved performance, achieving an AUC of 0.75 and an accuracy of 0.69 on the test set, both significantly higher than chance (*p*-value <0.001).

### Model interpretation

Figs 1 and 2 depict sample saliency maps for eight patients—two male and two female for each model class—using the ImageNet and sex-trained models, respectively. The original fundus images, Guided Grad-CAM outputs, and the colour-coded saliency maps are shown in the left, middle, and right panels, respectively. The images are selected as correctly classified examples from the test set. These sample maps show that the network focuses on distinct anatomical regions of the retina for male and female DR patients, with the highlights in male

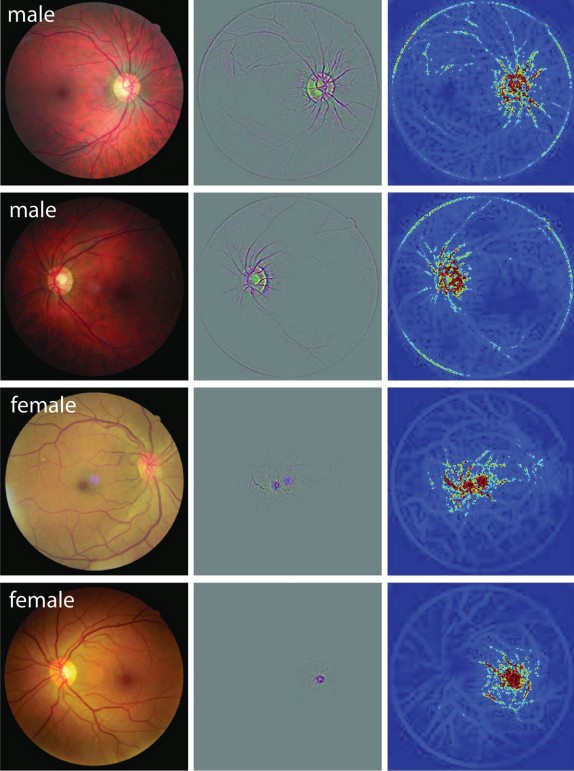

**Fig 1. Saliency map results of sample fundus images from two male and two female patients for the ImageNet pre-trained model.** In each panel, the original fundus image, the Guided Grad-CAM output (3-channel image), and its color-coded amplitude (single-channel image) are shown from left to right.

eyes consistently differing from those in female eyes. Specifically, the heatmaps for female eyes with DR predominantly highlight the macular region while largely ignoring the optic disc and peripheral regions. Conversely, heatmaps for male eyes with DR focused on the optic disc and peripheral vasculature along the arcades, with minimal emphasis on the central macular region. This pattern was consistently observed across the remaining correctly identified images in the test set (S1 and S2 Figs), raising the hypothesis that the information needed for sex classification based on fundus photographs in the presence of DR patients is located in different regions of the retina.

## Discussion

Here, we show that sex can be successfully predicted from fundus images of patients with DR—an attribute that psychophysical evidence indicates is undetectable by ophthalmologists above chance levels [27]—by fine-tuning pre-trained CNNs. The performance of machine learning models, especially deep neural networks, largely depends on the size of the training dataset. Given this dependence, our study achieved strong performance in sex classification from fundus photographs, particularly considering the small size of our training set, compared to previous studies that used much larger datasets [39,40]. As shown in Table 4, the performance we achieved in the present study is on par with, and even higher than what was achieved by previous work that used small datasets [27,38], despite using a smaller number of training samples. This may suggest that, in patients with DR, the retina may exhibit additional

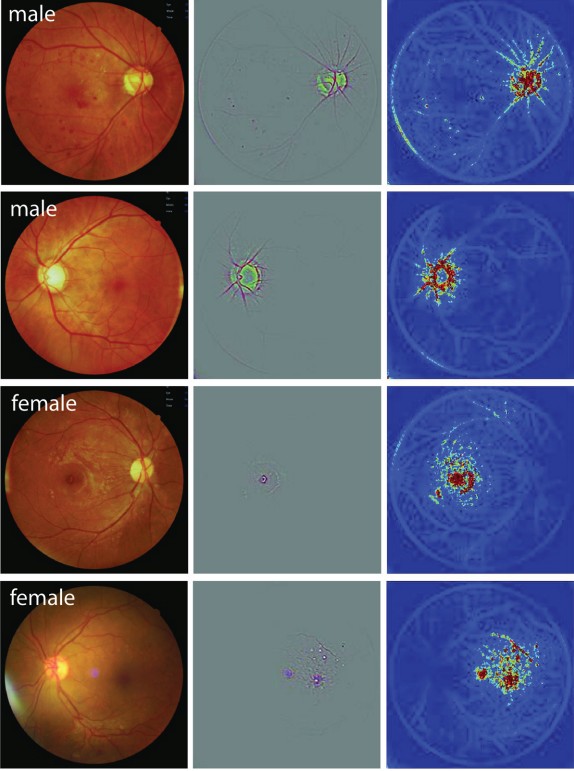

**Fig 2. Saliency map results of sample fundus images from two male and two female patients for the sex-trained model.** In each panel, the original fundus image, the Guided Grad-CAM output (3-channel image), and its colour-coded amplitude (single-channel image) are shown from left to right.

**Table 3. Model performance on the test set. AUCs and accuracies with their corresponding confidence intervals and p-values are reported. An asterisk (\*) denotes statistical significant results.**

|  | AUC ($CI_\alpha$) | p-value | Accuracy ($CI_\alpha$) | p-value |
|---|---|---|---|---|
| pre-trained on ImageNet | 0.718 (0.668, 0.769) | **< .001**\* | 0.663 (0.621, 0.708) | **< .001**\* |
| pre-trained on sex classification | 0.745 (0.700, 0.788) | **< .001**\* | 0.692 (0.650, 0.737) | **< .001**\* |

**Table 4. Sex classification results of the previous studies and the current study.**

|  | Training set images | AUC | $CI_\alpha$ |
|---|---|---|---|
| Poplin et al. [40] | 1,779,020 | 0.97 | (0.96, 0.98) |
| Korot et al. [39] | 173,819 | 0.93 | – |
| Delavari et al. [27] | 3,306 | 0.73 | (0.67, 0.79) |
| Berk et al.a [38] | 2,170 | 0.60 | (0.54, 0.65) |
| **Current work** | 2,071 | 0.75 | (0.70, 0.79) |
| Berk et al.b [38] | 1,746 | 0.71 | (0.66, 0.77) |

sex-specific features beyond what is known to differ in healthy eyes, potentially contributing to the improved predictive power of CNNs trained on eyes with DR. Importantly, the primary aim of this study was not sex classification from fundus images, but rather leveraging the trained model to investigate potential sex differences in the manifestation of DR.

In addition, post-hoc interpretation of the trained model shows that it focuses on different regions in male and female eyes. This is in stark contrast to saliency maps generated for healthy eyes [27], where the highlighted areas were similar for both male and female eyes: the optic disc, vasculature, and sometimes the macula. This focus on the same structures may suggest that sex differences in healthy eyes may involve variations in the same retinal structure (e.g. optic disc). However, in our study of DR-affected eyes, we observed distinct saliency map patterns for males and females: the model focused on the optic disc and the peripheral vasculature for males, avoiding the macular area, while for females, it concentrated on the macula with no focus on the optic disc or periphery. This was true for both of our models, including the one fine-tuned on a model already trained for sex classification on healthy eyes. Based on this, we hypothesize that sex differences in DR-affected eyes may be reflected in the model's attention to different retinal regions, though the specific features driving this behaviour remain unclear.

DR, from its onset at mild non-proliferative stages through moderate and severe non-proliferative DR to PDR and macular edema, manifests with hallmark lesions such as hard exudates, cotton wool spots, microaneurysms, and hemorrhages. Based on the interpretation of our models trained on eyes with DR, it is possible that the manifestation of DR differs between males and females. For instance, saliency maps generated by our model suggest that the macular region may play a more significant role in DR classification for females, whereas peripheral regions, particularly around the optic disc and along the vascular arcades, are more relevant for males. These patterns raise the hypothesis that women might be at a higher risk for macular edema, while men might be at a higher risk for PDR. However, it is important to note that saliency maps only highlight the regions the model attends to when making its predictions; while they inform us about which areas are important for the classification task, they do not definitively indicate which features or lesions, such as neovascularization or macular edema, are being used by the model. Further research is needed to validate this hypothesis; if confirmed, it could have significant implications for the diagnosis, management, and treatment of DR.

Differences in the manifestation of DR between males and females have not been extensively investigated, and our findings are among the first to suggest that such differences may exist. Failing to account for these differences in diagnostic approaches may lead to higher false positive and false negative rates. For example, diagnostic algorithms or clinical assessments that consider all symptoms (i.e., the union of symptoms in both sexes) may result in increased false positives, while those that focus only on common symptoms could overlook sex-specific features, increasing false negatives. By identifying and incorporating sex-specific features, diagnostic methods could be refined to reduce both types of errors, enable more accurate decisions, and mitigate the risks of delayed diagnosis by facilitating earlier detection of DR onset.

The risks of delayed diagnosis of DR, the leading cause of blindness among persons of working age in the industrialized world, are substantial and include progression to severe vision-threatening stages and systemic complications [44]. Early diagnosis alerts patients to the systemic onset of microvascular damage, signaling that their diabetes has progressed to a critical stage [45]. This awareness often prompts patients to adopt stricter glycemic control, especially for those who have not yet prioritized diabetes management. DR is also strongly correlated with diabetic kidney disease (DKD), and an early diagnosis of DR encourages closer monitoring of kidney function, including routine blood work, which can help detect and address kidney complications before they progress. Improved diabetes management and regular monitoring can reduce the risk of future complications, including cardiovascular disease, neuropathy, and kidney failure [46].

From an ocular perspective, early diagnosis is particularly beneficial for patients in the mild and moderate stages of DR, as the disease often progresses silently without symptoms in its early stages, delaying intervention. Regular follow-ups and monitoring at these stages provide an opportunity for timely intervention, which can help prevent progression to advanced stages [47]. More importantly, it is particularly crucial to identify and manage high-risk non-proliferative DR patients due to their approximately 50% likelihood of progressing to proliferative DR (PDR) within a year [48]. PDR, the advanced stage of DR, can cause blindness through retinal neovascularization-related complications such as vitreous hemorrhages, tractional retinal detachments, and neovascular glaucoma. At this advanced stage, even with treatment, the likelihood of meaningful visual recovery is significantly reduced [49].

Current treatment options for DR, including pan-retinal photocoagulation (PRP) and intravitreal anti-VEGF and glucocorticosteroid injections, have revolutionized outcomes for DR patients [47]. Severe vision loss in DR was once common, affecting approximately 50% of patients with PDR in the pre-treatment era [50]. Today, these rates have decreased in patients receiving appropriate treatment. However, the effectiveness of these treatments is closely tied to the stage of DR at the time of intervention. The chance of preventing vision loss or achieving visual recovery diminishes significantly as the disease progresses to advanced stages [51]. Thus, early and more accurate diagnosis not only reduces diagnostic errors but also enables timely intervention, which can prevent the progression of DR and significantly improve patient outcomes.

Our analysis was based on the binary 'patient-sex' field in the dataset, which classifies individuals as either male or female. The observed differences may reflect biological aspects of sex, such as hormonal or anatomical variations, or may be influenced by social factors commonly associated with gender, including healthcare access and lifestyle behaviours. However, as our dataset lacks sociocultural and behavioural information, we are unable to disentangle these influences. Future studies incorporating more gender-specific data will be essential to better understand the distinct contributions of sex and gender to DR manifestation.

Another limitation of our study is the absence of healthy control images in the primary analysis. The EyePACS dataset used in this study includes only individuals with some degree of diabetic retinopathy, preventing direct comparison with healthy eyes. While previous work has investigated sex-based differences in healthy fundus images, we did not include a healthy cohort in our model training or evaluation. Incorporating a healthy control group from a different dataset would introduce data distribution shifts due to differences in imaging protocols, equipment, or population characteristics. Such shifts could confound the model's behaviour and make it hard to isolate effects specifically related to DR. Future studies using harmonized datasets that include both healthy and DR-affected eyes from the same source would be valuable in disentangling general sex-based retinal differences from those specific to disease manifestation.

In addition to these directions, future work could benefit from external validation on independent datasets such as Messidor [52] or IDRiD [53] to assess the generalizability of our findings across different populations and imaging conditions. Moreover, although the EyePACS dataset includes individuals from diverse ethnic backgrounds, we did not explore whether sex-related differences in retinal presentation vary across ethnic groups. Understanding how sex and ethnicity may interact in the context of DR could offer valuable insights into disease manifestation and the model's behaviour, and represents an important avenue for future research.

Finally, this study serves as a proof-of-concept, demonstrating the potential of deep learning-based analysis of fundus images to uncover novel retinal biomarkers, enabling more precise diagnosis and management of retinal diseases. Our findings suggest that DR may

manifest differently in males and females, highlighting the need for sex-specific diagnostic approaches. While our model successfully identified sex-based retinal differences, further research is required to validate these findings in larger, more diverse datasets and to determine their clinical significance. Ultimately, leveraging AI-driven insights could help refine clinical decision-making, reduce diagnostic errors, and improve outcomes for patients with DR.

## Supporting information

**S1 Table. Composition of the CNN Training set.**
(PDF)

**S2 Table. Composition of the CNN Validation set.**
(PDF)

**S3 Table. Composition of the CNN Test set.**
(PDF)

**S1 Fig. Additional saliency map results for the model pre-trained on ImageNet.** Sixteen images were randomly chosen to demonstrate the consistency of the saliency map results.
(TIF)

**S2 Fig. Additional saliency map results for the model pre-trained on sex classification.** Sixteen images were randomly chosen to demonstrate the consistency of the saliency map results.
(TIF)

## Author contributions

**Conceptualization:** Parsa Delavari, Gulcenur Ozturan, Eduardo V. Navajas, Ozgur Yilmaz, Ipek Oruc.

**Data curation:** Gulcenur Ozturan.

**Formal analysis:** Parsa Delavari.

**Funding acquisition:** Ozgur Yilmaz, Ipek Oruc.

**Investigation:** Parsa Delavari, Gulcenur Ozturan, Eduardo V. Navajas, Ozgur Yilmaz, Ipek Oruc.

**Methodology:** Parsa Delavari, Gulcenur Ozturan, Eduardo V. Navajas, Ozgur Yilmaz, Ipek Oruc.

**Project administration:** Ozgur Yilmaz, Ipek Oruc.

**Resources:** Ozgur Yilmaz, Ipek Oruc.

**Software:** Parsa Delavari, Gulcenur Ozturan, Eduardo V. Navajas, Ozgur Yilmaz, Ipek Oruc.

**Supervision:** Ozgur Yilmaz, Ipek Oruc.

**Validation:** Parsa Delavari, Gulcenur Ozturan, Eduardo V. Navajas, Ozgur Yilmaz, Ipek Oruc.

**Visualization:** Parsa Delavari, Gulcenur Ozturan, Eduardo V. Navajas, Ozgur Yilmaz, Ipek Oruc.

**Writing – original draft:** Parsa Delavari.

**Writing – review & editing:** Parsa Delavari, Gulcenur Ozturan, Eduardo V. Navajas, Ozgur Yilmaz, Ipek Oruc.

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
