## [Editor Report · Decision Letter 0]

23 Sep 2024

PONE-D-24-39869

AI-Assisted identification of sex-specific patterns in diabetic retinopathy using retinal fundus images

PLOS ONE

Dear Dr. Delavari,

Thank you for submitting your manuscript to PLOS ONE. After careful consideration, we have decided that your manuscript does not meet our criteria for publication and must therefore be rejected.

Specifically:

The manuscript is concerned with the application of the VGG16 architecture for analysis of fundus images in the context of gender-specific diabetic retinopathy, coupled with Guided Grad-CAM to support the interpretability of the results, through the analysis of the associated feature maps. The application area of this research is interesting, however, there are a few major concerns.Overall, beyond the application area, there is a lack of novelty as both techniques are very well known in the state-of-art.VGG16 is relatively shallow, which may limit its ability to capture the complex patterns and subtle features that are critical for detecting early signs of diabetic retinopathy, especially when factoring in gender-specific variations. Moreover, it employs a series of max-pooling layers that downsample the input images progressively. This aggressive downsampling can lead to a loss of fine spatial details that are crucial for detecting small lesions, microaneurysms, or subtle gender-specific retinal features associated with diabetic retinopathy.If VGG16 misclassifies a fundus image due to its inherent limitations (e.g., poor handling of small-scale features), Guided Grad-CAM will still produce a heatmap, but the highlighted areas may be misleading, pointing to regions that have no relation to diabetic retinopathy or gender-specific features. This is problematic in medical contexts, where misinterpretation can affect diagnosis and treatment decisions.Importantly, the results of Guided Grad-CAM are affected by the choice of the CNN structure. Since Guided Grad-CAM generates visual explanations by analyzing gradients flowing through the network and highlighting important regions based on the activation maps of the convolutional layers, the CNN architecture plays a critical role in shaping the output.Instead, additional experiments with a vareity of CNN architectures are needed, or alternatively, to consider approaches such as attention mechanisms, self-supervised learning and integrated gradients.

I am sorry that we cannot be more positive on this occasion, but hope that you appreciate the reasons for this decision.

Kind regards,

Panos Liatsis, PhD

Academic Editor

PLOS ONE

- - - - -

---

## [Author Response · Author response to Decision Letter 1]

7 Oct 2024

Our manuscript was not sent out for peer-review, therefore there are no reviewer comments that we can respond to. The response letter to the Academic Editor's comments in a point-by-point fashion is uploaded as a PDF file.

---

## [Decision Letter · Decision Letter 1]

23 Dec 2024

PONE-D-24-39869R1AI-Assisted identification of sex-specific patterns in diabetic retinopathy using retinal fundus imagesPLOS ONE

Dear Dr. Delavari,

Thank you for submitting your manuscript to PLOS ONE. After careful consideration, we feel that it has merit but does not fully meet PLOS ONE’s publication criteria as it currently stands. Therefore, we invite you to submit a revised version of the manuscript that addresses the points raised during the review process.

The manuscript has been evaluated by two reviewers, and their comments are available below.

The reviewers have raised a number of major concerns. They request improvements to the reporting of methodological aspects of the study, clarification on the interpretation of the results and further discussion.

Could you please carefully revise the manuscript to address all comments raised?

We also note that one or more reviewers has recommended that you cite specific previously published works. As always, we recommend that you please review and evaluate the requested works to determine whether they are relevant and should be cited. It is not a requirement to cite these works. We appreciate your attention to this request.

We look forward to receiving your revised manuscript.

Kind regards,

Helen Howard

Staff Editor

PLOS ONE

Journal Requirements:

3. Please expand the acronym “NSERC” (as indicated in your financial disclosure) so that it states the name of your funders in full.

Additional Editor Comments (if provided):

Reviewers' comments:

Reviewer's Responses to Questions

**Comments to the Author**

1. If the authors have adequately addressed your comments raised in a previous round of review and you feel that this manuscript is now acceptable for publication, you may indicate that here to bypass the “Comments to the Author” section, enter your conflict of interest statement in the “Confidential to Editor” section, and submit your "Accept" recommendation.

Reviewer #1: All comments have been addressed

Reviewer #2: (No Response)

2. Is the manuscript technically sound, and do the data support the conclusions?

Reviewer #1: Yes

Reviewer #2: Partly

3. Has the statistical analysis been performed appropriately and rigorously? 

Reviewer #1: Yes

Reviewer #2: No

4. Have the authors made all data underlying the findings in their manuscript fully available?

Reviewer #1: No

Reviewer #2: Yes

5. Is the manuscript presented in an intelligible fashion and written in standard English?

Reviewer #1: Yes

Reviewer #2: Yes

6. Review Comments to the Author

Reviewer #1: • The rationale for identifying sex-specific patterns in diabetic retinopathy is not well-justified. The authors should provide a stronger explanation of the clinical relevance and potential impact of their findings.

• It is unclear how this approach contributes to improving diagnosis or treatment strategies compared to traditional methods.

• The dataset used for training and validation is not described in detail. Critical information such as sample size, class balance, preprocessing steps, and demographic distribution is missing.

• Include more recent references that are relevant to the topic, such as

• Al-hazaimeh, Obaida M., Ashraf A. Abu-Ein, Nedal M. Tahat, Ma'moun A. Al-Smadi, and Malek M. Al-Nawashi. "Combining Artificial Intelligence and Image Processing for Diagnosing Diabetic Retinopathy in Retinal Fundus Images." International Journal of Online & Biomedical Engineering 18, no. 13 (2022).

• Gharaibeh, Nasr, Obaida M. Al-Hazaimeh, Bassam Al-Naami, and Khalid MO Nahar. "An effective image processing method for detection of diabetic retinopathy diseases from retinal fundus images." International Journal of Signal and Imaging Systems Engineering 11, no. 4 (2018): 206-216.

Reviewer #2: It is a good study with a reasonably good sample size of retinal images, which is age and gender adjusted.

1. While the conclusions mentions that diabetic macular edema (DME) appears to be more common in women and PDR in men as assessed by the AI system, I do not find any information on DME in the tables supporting the conclusion. Can you kindly provide the same?

2. If the duration of diabetes is available, that information can be added as it is one of the key factors in DR irrespective of the gender

3. Kindly add in the discussion the advantages of knowing the gender based on the use of AI with retinal images.

How does it help in planning screening / management of DR? These points have to be there to support the study in the discussion section

7. PLOS authors have the option to publish the peer review history of their article (what does this mean?). If published, this will include your full peer review and any attached files.

Reviewer #1: No

Reviewer #2: No

---

## [Author Response · Author response to Decision Letter 2]

14 Feb 2025

In the "Response to Reviewers" letter, we have responded to the editor comments and addressed each issue raised by the reviewers.

---

## [Decision Letter · Decision Letter 2]

25 Apr 2025

PONE-D-24-39869R2AI-Assisted identification of sex-specific patterns in diabetic retinopathy using retinal fundus imagesPLOS ONE

Dear Dr. Delavari,

Thank you for submitting your manuscript to PLOS ONE. After careful consideration, we feel that it has merit but does not fully meet PLOS ONE’s publication criteria as it currently stands. Therefore, we invite you to submit a revised version of the manuscript that addresses the points raised during the review process.

**Few comments:**

**- The discussion suggests females may be more prone to ME and males to PDR, but the model only identifies regions of attention, not pathology. There are details comments from this from one of the reviewers that need to be addressed for the manuscript to be accepted.**

**- For future research, have you consider validation on an external dataset (e.g., Messidor or IDRiD) to enhance generalizability. This could be ackonwledged in the discussion.**

**- Though the dataset is stratified by ethnicity, the discussion does not explore whether observed sex-specific differences vary across ethnic groups. This is an opportunity for future research but should be acknowledged.**

We look forward to receiving your revised manuscript.

Kind regards,

Tomo Popovic, Ph.D.

Academic Editor

PLOS ONE

**Journal Requirements:**

Reviewers' comments:

Reviewer's Responses to Questions

**Comments to the Author**

1. If the authors have adequately addressed your comments raised in a previous round of review and you feel that this manuscript is now acceptable for publication, you may indicate that here to bypass the “Comments to the Author” section, enter your conflict of interest statement in the “Confidential to Editor” section, and submit your "Accept" recommendation.

Reviewer #2: All comments have been addressed

Reviewer #3: (No Response)

2. Is the manuscript technically sound, and do the data support the conclusions?

Reviewer #2: Yes

Reviewer #3: Partly

3. Has the statistical analysis been performed appropriately and rigorously? 

Reviewer #2: Yes

Reviewer #3: Yes

4. Have the authors made all data underlying the findings in their manuscript fully available?

Reviewer #2: Yes

Reviewer #3: No

5. Is the manuscript presented in an intelligible fashion and written in standard English?

Reviewer #2: Yes

Reviewer #3: Yes

6. Review Comments to the Author

**Reviewer #2: **the comments have been addressed satisfactorily by the authors. The revised manuscript reads well. showing sex differences in DME AND PDR

**Reviewer #3: **This manuscript explores the important and timely topic of sex-specific differences in diabetic retinopathy (DR), a complication of diabetes that remains a major cause of vision impairment worldwide. To investigate this, the authors employ an original and creative approach—training convolutional neural networks (CNNs) to classify fundus images as either male or female in patients with DR. The underlying hypothesis is that if DR manifests differently by sex, those differences would be recognized by the CNN as distinguishing features, and subsequently highlighted in saliency (heat) maps.

This manuscript addresses a novel and clinically relevant question using innovative AI-based methodology. The work is technically sound and the results are intriguing.

However, I recommend revisions to ensure consistency in terminology, more cautious interpretation of findings, and a clearer discussion of the study’s limitations. With these adjustments, the paper will make a meaningful contribution to the growing field of AI in ophthalmology and precision medicine.

Comments:

1. Clarification of Sex vs. Gender Terminology: The manuscript would benefit from a more consistent and accurate use of terminology. Sex is a biological classification based on physiological and anatomical characteristics (e.g., chromosomes, hormone profiles, reproductive anatomy), and is indeed relevant to metabolic processes and disease manifestations—as is the focus of this study. In contrast, gender refers to social and cultural roles, personal identity, and lived experience, and is relevant to factors such as access to healthcare, health-seeking behavior, and exposure to stigma or stress—all of which can also impact health outcomes.

However, because the study compares only two groups—male and female—based solely on retinal morphology, and does not include gender identity or sociocultural data, it is more accurate to frame the findings strictly in terms of sex differences. Including both sex and gender as interchangeable terms risks introducing conceptual ambiguity and potentially overextends the claims.

For clarity and scientific precision, I recommend that the manuscript use the term sex consistently throughout and revise or remove references to gender unless supported by appropriate demographic or behavioral data. This will improve the coherence and readability of the text and avoid conflating two distinct dimensions of identity.

2. Scope of Analysis and Grouping Strategy: If the intent were to explore both sex and gender differences, the analysis would require more than two comparison groups (e.g., accounting for intersex, transgender, or non-binary individuals), which would significantly increase complexity and demand an entirely different dataset. Since the current study includes only biologically classified males and females, it is appropriate to keep the analysis focused accordingly. Clarifying this in the introduction and discussion will help anchor the scientific message and prevent misinterpretation.

3. Absence of Healthy Controls: One notable limitation, which should be more explicitly acknowledged, is the lack of healthy control images in the primary analysis. While the authors cite previous work involving healthy eyes, this study exclusively includes DR-affected eyes. Without healthy controls, it is difficult to determine whether the CNN is identifying sex-specific differences in DR pathology or simply general sex-based anatomical differences that persist regardless of disease. Addressing this limitation more directly in the discussion would enhance the manuscript’s rigor.

4. Interpretation of AI Output and Causality: The strongest concern lies in the interpretation of the saliency maps and AI-derived findings. Saliency maps indicate the regions of the image that influenced the model’s classification decision, but they do not reveal which specific features or pathological changes were used in that decision. Thus, while the CNNs successfully distinguish male from female fundus images with moderate accuracy, this does not confirm the presence of sex-specific disease patterns in DR.

For example, the observed focus on the macular region in female eyes does not necessarily indicate an increased risk of macular edema in women. At most, the findings support that the macula in diabetic females has a distinct appearance compared to diabetic males. To establish a causal link between these imaging features and specific clinical outcomes (e.g., macular edema or progression to PDR), longitudinal studies would be required. This paper does not provide such outcome data or analysis and should avoid implying it does.

Therefore, the current claims regarding differential risk (e.g., "females may be at higher risk for macular edema") are speculative and should be clearly presented as hypotheses for future study, not as conclusions drawn from the present data.

5. Recommendation on Framing and Language: The manuscript would benefit from more cautious phrasing when presenting its conclusions. Statements such as "these findings suggest..." should be tempered with language like "these findings raise the hypothesis that..." or "these findings may indicate...". This will align better with the nature of the data and the exploratory design of the study.

7. PLOS authors have the option to publish the peer review history of their article (what does this mean?). If published, this will include your full peer review and any attached files.

Reviewer #2: No

Reviewer #3: No

---

## [Author Response · Author response to Decision Letter 3]

6 Jun 2025

We have uploaded a Response Letter in which we have addressed the comments raised by the editor and the reviewers.

---

## [Editor Report · Decision Letter 3]

13 Jun 2025

AI-Assisted identification of sex-specific patterns in diabetic retinopathy using retinal fundus images

PONE-D-24-39869R3

Dear Dr. Delavari,

We’re pleased to inform you that your manuscript has been judged scientifically suitable for publication and will be formally accepted for publication once it meets all outstanding technical requirements.

Kind regards,

Tomo Popovic, Ph.D.

Academic Editor

PLOS ONE
---

## [Editor Report · Acceptance letter]

PONE-D-24-39869R3

PLOS ONE

Dear Dr. Delavari,

I'm pleased to inform you that your manuscript has been deemed suitable for publication in PLOS ONE. Congratulations! Your manuscript is now being handed over to our production team.

Kind regards,

on behalf of

Prof. Tomo Popovic

Academic Editor

PLOS ONE